# Barriers to exclusive breastfeeding practice among HIV-positive mothers in Tanzania. An exploratory qualitative study

**Goodluck Augustino** *, **Amani Anaeli**, **Bruno F. Sunguya**

School of Public Health and Social Sciences, Muhimbili University of Health and Allied Sciences, Dar Es Salaam, Tanzania

* goodluckaugustino@gmail.com

**Data Availability Statement:** All relevant data are within the manuscript and its Supporting Information files.

## Abstract

### Background

Ensuring optimal nutrition through early breastfeeding is vital for infant mental development and overall health. HIV infections complicate decisions regarding exclusive breastfeeding, jeopardizing effective infant and young child feeding, which affects nutrition and health outcomes. Recognizing the lack of evidence on barriers to infant feeding in the context of HIV in Tanzania, this study was conducted to explore individual, household, and community obstacles in the Ilala district, Dar es Salaam.

### Materials and methods

A case study design employing a qualitative approach was used. The study was executed at the Reproductive and Child Health (RCH) Clinic within Amana Regional Referral Hospital, Mnazi Mmoja Health Centre, and Buguruni Health Centre. Data collection ensued through the purposive sampling of healthcare providers and HIV-positive mothers, utilizing in-depth interview techniques. The textual data accrued were analyzed using inductive and deductive content analysis strategies, thereby enabling the delineation and formulation of principal thematic constructs.

### Results

The study involved interviews with twenty-seven key informants, encompassing HIV-positive mothers, nurses, clinicians, and community-based healthcare workers. The analysis of these interviews resulted in the identification of three major themes. Firstly, individual barriers to exclusive breastfeeding were delineated, encompassing sub-themes such as occupation-related hectic schedules, early motherhood-related non-compliance to safe infant feeding directives, postpartum depression, and issues related to breast sores and abscesses. Secondly, household-level barriers were identified, highlighting challenges like food insecurity and inaccessibility to key resources, the influence of male partners and family members on decision-making, and barriers arising from non-disclosure of HIV status affecting Exclusive Breastfeeding (EBF) support. Lastly, community-level barriers were

**Funding:** Data collection wa supported by HIV Implementation Science Grant and NIH D43 grant (1D43TW009775-01A1). The funders had no role in study design, data collection and analysis, decision to publish, or preparation of the manuscript.

**Competing interests:** The authors have declared that no competing interests exist.

explored, revealing a sub-theme related to the low retentivity of HIV-positive women in the Prevention of Mother-to-Child Transmission (PMTCT) programs.

## Conclusions

Individual barriers encompass practical, psychological, and physical challenges, while household-level obstacles include food insecurity, limited resources, and family dynamics influencing decisions. At the community level, there's a concern about the low retentivity of HIV-positive women in PMTCT programs, indicating broader societal challenges in supporting exclusive breastfeeding. There is a need for tailored interventions at individual, household, and community levels to promote and support optimal infant feeding practices among HIV-positive women.

## Introduction

Optimal nutrition for infants is vital for their development, particularly during the critical first 24 months of life. The period sets the foundation for healthy growth and development, impacting future generations [1]. The World Health Organization (WHO) recommends Exclusive Breastfeeding (EBF) for the first six months, complementary feeding for up to 12 months, and continued breastfeeding for up to 24 months [2]. However, the estimated rate of EBF is 41% in Sub-Saharan Africa (SSA), with vast regional variations [3]. Despite an average breastfeeding duration of 14–19 months in SSA, only 30% - 46% practice EBF for the recommended six months [4–6]. In Tanzania, while the EBF rate has improved to 59%, challenges persist, making it crucial to understand the dynamics influencing breastfeeding practices [7].

HIV/AIDS has complicated infant and Young Child Feeding (IYCF) practices. Although the benefit of breastfeeding outweighs the risks even for children exposed to HIV, far fewer children exposed to the infections are exclusively breastfed in fear of new infections. In 2017, an estimated 360,000 children were newly infected with HIV globally, and a significant portion of these infections occurred during breastfeeding [8]. While the combination of ART and EBF has proven effective in reducing HIV transmission to less than 1% [9], challenges arise in maintaining consistent ART with EBF breastfeeding [10]. Unlike high-income countries where breastfeeding is discouraged postnatally, low and middle-income countries face the dilemma of unsafe formula feeding, leading to higher mortality rates in children under five [11].

In the context of HIV, adherence to the recommended duration of EBF remains a challenge [12]. Inconsistent EBF practices among HIV-positive mothers have been observed across SSA countries, reflecting the need for tailored intervention [13–16]. Tanzania, despite progress, faces barriers leading to varied EBF rates, such as the decline from 85% at two months to <30% by four months in some areas [17, 18]. Factors contributing to these challenges include socioeconomic, cultural, and health-related barriers, emphasizing the need for a comprehensive understanding of the contextual influences on breastfeeding practices [12, 17–21].

While various studies have explored factors influencing exclusive breastfeeding practices, limited attention has been given to understanding barriers specific to HIV-exposed infants in Tanzania. Existing research has primarily focused on economic aspects, leaving gaps in addressing cultural and societal norms impacting EBF [12, 21]. This study aims to fill these knowledge gaps by investigating barriers to EBF among HIV-positive mothers according to

WHO guidelines in Tanzania. By employing an integrated model of behavioral prediction [22], we aim to provide insights into the determinants affecting exclusive breastfeeding practices, paving the way for targeted interventions.

## Materials and methods

### Ethical consideration

This study received ethical approval from the Research Ethics Committee of Muhimbili University of Health and Allied Sciences (Ref. No. DA.282/298/01.C/MUHAS-REC-08-2021-813). Additionally, research clearance was obtained from the Dar es Salaam Regional Authority and Ilala Council Authority. Written consent was diligently obtained from all participants, accompanied by a clear assurance of their right to withdraw from the study at any point. Maintaining confidentiality and respecting privacy were paramount, with no names being recorded in the transcripts.

### Research design

This study adopted a case study design with a qualitative approach to explore barriers to exclusive breastfeeding practice among HIV-positive mothers. The selection of a case study design was deliberate, driven by the fact that the chosen diverse cases represented real, ongoing, and current scenarios [23]. Many controls on infant feeding choices are knotty and complex to quantify [24]. Therefore, using a qualitative method provided a more profound comprehension of the factors that impact infant feeding behaviors and intentions.

### Setting

The study was conducted in the Ilala district in Dar es Salaam. The region was chosen as it is Tanzania's largest city, industrial center, and major port [25]. In addition to the significant population with mixed cultures from all over and outside the country, the urban-rural migration status in Dar es Salaam is higher than in other regions in Tanzania [25, 26]. Further, the city is projected to grow beyond 10 million by 2029 and emerge among megacities [26]. Urbanization's influence on breastfeeding and HIV is a well-established [1, 27, 28].

The study was done in the highest client flow accessing HIV services through HIV Care and Treatment Centers (CTC) and Reproductive and Child Health (RCH) clinics in Ilala, Amana Regional Referral Hospital, Mnazi Mmoja, and Buguruni Health Centre. Although the regional proportion of EBF is not known among HIV-exposed infants, the overall estimate of EBF in Dar es Salaam among children in the general population is 57.1% [29].

### Recruitment and selection of study participants

Purposive sampling was used to recruit HIV-positive breastfeeding mothers, health facility-based healthcare providers (clinicians), and Community Health workers in the Ilala district in Dar es Salaam. HIV-positive mothers who had infants aged 3–6 months, regardless of their current feeding practice, have been placed on a treatment regimen (ARV medicines), attending CTC and RCH between March 08[th] to 25[th], 2022, recruitment period of the study were eligible. This age range was chosen for the study based on previous research findings that mothers in Tanzania typically discontinue breastfeeding or begin introducing complementary foods before or around the time their infants reach three months [7, 17, 30].

Further, a hospital document review was conducted to support a judgmental sampling procedure based on the diversity of sub-categories and to identify the participants who met the sampling criteria. These sub-categories include socio-demographic status (age–Adolescents,

above 18 years, educated (at least standard 7 of primary school), uneducated, parity (first child-parent, multiple children), marital status (polygyny, Monogamy, single mother), infant's gender, and employment status (employed, unemployed) as they have been proven to impact exclusive breastfeeding practice in women in the general population [31]. Additionally, clinicians from CTC and RCH of the selected health facilities and their community health workers were purposively recruited. Study staff approached eligible healthcare providers and HIV-positive mothers in person at the RCH clinics and asked for an interview after briefing them about the study. Twenty-seven interviews were adequate to reach saturation of the data collected in the selected health facilities, including twenty-one in-depth interviews with HIV-positive mothers, four healthcare providers, and two community healthcare workers key informant interviews. All eligible participants approached agreed to participate, and no repeat interviews were conducted.

## Data collection and tools

Each group was assigned a precise in-depth interview (IDI) guide (see S1 File). These guides underwent a pilot test involving a subset of patients, a healthcare provider, and a community healthcare worker, leading to the refinement of questions. Each guide consisted of open-ended questions accompanied by probing queries, specifically tailored to address the three study objectives focusing on Barriers to Exclusive Breastfeeding Practices Among HIV-Positive Women in Dar Es Salaam.

The in-depth interviews, aligned with methodologies in previous Sub-Saharan African studies [32], delved into personal aspects such as reasons, feelings, beliefs, and opinions surrounding infant feeding within the context of HIV [33]. This approach provided rich insight, given the deeply rooted connections to cultural and social norms [34]. Complementing these in-depth interviews, key informant interviews involved healthcare providers from selected clinics and community health workers. These informants were chosen for their perceived information richness concerning exclusive breastfeeding practices among HIV-positive women.

The interviews were conducted by SJ, the researcher, a female medical doctor with an MPH and an affiliation with a consulting research firm in Tanzania. SJ has undergone extensive training in qualitative research methods and has actively participated in qualitative research for over two years. The interviews took place in a dedicated and isolated room, carefully chosen to avoid the presence of healthcare providers and other patients. This intentional measure aimed to safeguard key informants' privacy and maintain the shared information's confidentiality. SJ (MD, MPH) also thoroughly explored each question with participants, concluding data collection when all identified themes became saturated.

The interviews were carried out in Kiswahili, the national language. Twenty-seven in-depth interviews were conducted, each audio-recorded using Olympus WS-852, after the participants' written informed consent. The details of informed consent, encompassing confidentiality, audio recording, and the anonymous dissemination of results, were thoroughly explained at the commencement of each interview. Once consent was obtained and documented, the interview and audio recording commenced. To protect participants' identities, study numbers were assigned to each individual. The interview duration ranged from 20 to 40 minutes. Additionally, field notes were recorded for each interview to aid in the analysis process.

## Data management and analysis strategies

The data were subjected to content analysis using deductive and inductive approaches, a method chosen to mitigate potential weaknesses associated with employing a single approach

[35]. The researcher transcribed the audio-taped data verbatim into a Microsoft Word document using the Express Scribe computer program. Bilingual expertise facilitated the translation of all transcripts from Swahili to English. Thorough readings of the transcripts were conducted to comprehend key informants' portrayed concepts, which were then coded using predetermined and emergent codes. Predetermined codes were derived from existing studies on barriers to exclusive breastfeeding among HIV-positive mothers [32, 36–40], while new codes were formulated for information not fitting within these categories. The codes, both predetermined and emergent, were organized into nine clusters, termed sub-themes, which were further condensed into three main themes: Individual Barriers to Exclusive Breastfeeding, Household-Level Barriers to Exclusive Breastfeeding, and Community-Level Barriers to Exclusive Breastfeeding. Notably, the sub-themes "Early motherhood-related non-compliance to safe infant feeding directives," "HIV status non-disclosure barriers to EBF support," and "Low retentivity of HIV-positive women in the PMTCT program" were derived directly from the analyzed text data rather than existing literature. The iterative nature of data collection and analysis highlighted the robustness of inductive and deductive content analysis methods. The systematic assignment of texts into codes into sub-themes and sub-themes into main themes was facilitated by modifying the coding scheme when new information emerged. The extensive volume of text data was managed using NVivo version 12 computer software.

## Results

### Description of study participants

The study involved six healthcare providers, one male and five females aged between 30 and 48 years. Among the PMTCT clients interviewed, 48% of HIV-positive mothers were in the 25–34 age range. Notably, 57% of participants had their index child as the 2nd or 3rd, while 41% were first-time mothers. Most (67%) participants had education levels around the primary level, and 52% reported involvement in small businesses. In contrast, 83% of healthcare providers had at least five years of experience in PMTCT, with one having two years in CTC and PMTCT. Among the six providers, 50% had diplomas, and all played a crucial role as key informants, offering diverse perspectives on barriers to exclusive breastfeeding among HIV-positive women (Table 1).

### Presentation of findings

Three themes emerged, and eight sub-themes were constructed from the data, and these are summarized in Table 2 below.

**Theme 1: Individual barriers to exclusive breastfeeding.** Participants in this study described individual barriers faced during exclusive breastfeeding. Under this theme, four sub-themes emerged, including early motherhood-related non-compliance to the safe infant feeding directives, occupation-related schedule, postpartum depression, and breast sores and abscesses. The sub-themes are detailed below with support of verbatim quotations.

*Occupation-related schedule.* Our study reveals that strict rules of mothers' occupation have significantly influenced mixed feeding. The work that capacitated them to provide for themselves from the income generated made mothers trade off caring for their infants over their jobs. As a result, they neither offer maternity leave, mid-day time out to breastfeed nor reduced work hours to breastfeed their infants, and self-caring to produce enough milk for their infants. One of the participants, who shared their valuable insights, highlighted a challenging situation. This participant said:

> *"I had to stop my child from breastfeeding and leave him with soft porridge since I am working as a maid, where we work 12-hour shifts." (P12, Health Center)*

**Table 1. Socio-demographic findings of the study participants.**

| SN | Sex | Age | Education | Marital Status | Number of Children | Occupational Status | Participant identity in this research |
|---|---|---|---|---|---|---|---|
| 1 | F | 25 | Primary | Married | 2 | Other | P1 Health Facility 1 |
| 2 | F | 32 | Primary | Separated | 3 | Employed | P2 Health Facility 1 |
| 3 | F | 26 | Primary | Married | 2 | Business | P3 Health Facility 1 |
| 4 | F | 27 | Secondary | Married | 2 | Business | P4 Health Facility 1 |
| 5 | F | 27 | Secondary | Married | 2 | Employed | P5 Health Facility 1 |
| 6 | F | 29 | Secondary | Not Married | 2 | Business | P6 Health Facility 1 |
| 7 | F | 25 | Secondary | Not Married | 1 | Employed | P7 Health Facility 1 |
| 8 | F | 33 | Secondary | Married | 3 | Employed | P8 Health Facility 1 |
| 9 | F | 34 | College | Married | 4 | Business | P9 Health Facility 1 |
| 10 | F | 30 | Secondary | Married | 2 | Business | P1 Health Facility 2 |
| 11 | F | 19 | Secondary | Not Married | 1 | Other | P2 Health Facility 2 |
| 12 | F | 18 | Primary | Not Married | 1 | Other | P3 Health Facility 2 |
| 13 | F | 20 | Secondary | Separated | 1 | Business | P4 Health Facility 2 |
| 14 | F | 22 | Secondary | Separated | 1 | Business | P5 Health Facility 2 |
| 15 | F | 21 | Secondary | Not Married | 1 | Employed | P6 Health Facility 2 |
| 16 | F | 24 | Secondary | Not Married | 1 | Business | P7 Health Facility 2 |
| 17 | F | 40 | Primary | Separated | 3 | Other | P1 Health Facility 3 |
| 18 | F | 35 | Primary | Married | 2 | Business | P2 Health Facility 3 |
| 19 | F | 35 | Secondary | Married | 2 | Business | P3 Health Facility 3 |
| 20 | F | 36 | Secondary | Married | 3 | Employed | P4 Health Facility 3 |
| 21 | F | 36 | Secondary | Married | 4 | Business | P5 Health Facility 3 |
| 22 | M | 48 | College | | | Healthcare provider | Provider 1, Health Facility 1 |
| 23 | F | 33 | Secondary | | | Healthcare provider | Provider 2, Health Facility 1 |
| 24 | F | 40 | College | | | Healthcare provider | Provider, Health Facility 2 |
| 25 | F | 32 | College | | | Healthcare provider | Provider, Health Facility 3 |
| 26 | F | 33 | Primary | | | Community Health worker | CHW, Ward 1 |
| 27 | F | 30 | Primary | | | Community Health worker | CHW, Ward 2 |

This statement underscores the impact of employment on exclusive breastfeeding practices. The participant's work schedule and responsibilities made it logistically challenging to breast-feed their child during mid-day. Furthermore, the participant expressed concerns about potential risks associated with the child's caretaker. She explained the following:

**Table 2. Theme and sub-themes.**

| Theme number | Themes | Sub-themes |
|---|---|---|
| Theme 1 | Individual barriers to exclusive breastfeeding | Occupation-related hectic schedule |
| | | Early motherhood-related non-compliance to the safe infant feeding directives |
| | | Postpartum depression |
| | | Breast sores and abscess |
| Theme 2 | Household-level barriers to exclusive breastfeeding | Food insecurity and inaccessibility to key resources |
| | | Male partners and family members influence on decision making |
| | | HIV status non-disclosure barriers to EBF support |
| Theme 3 | Community-level barriers to exclusive breastfeeding | Low retentivity of HIV-positive women in PMTCT programs |

*"Moreover, I was afraid that when I was not around, my young sister, whom I left the baby with, might be tempted to give him sweet drinks like Soda to calm him; when my child gets irritated and crying, that might be more dangerous." (P12, Health Center)*

This apprehension indicates the awareness of the importance of exclusive breastfeeding and the potential consequences of introducing non-breast milk items to the infant's diet.

We also gained valuable insights from a community health worker (CHW) behind mixed feeding practices among mothers, which are often influenced by the mothers' occupations, particularly those working in industry-related jobs and Hindu facilities. The main challenge for these women is the need for more time off to breastfeed their infants frequently. The participant said:

*The most reason we encounter when trying to probe mixed feeding practice is influenced by their occupation, which comes from these industry-related occupations and those working in Hindu facilities not being given time off to breastfeed their kids. Moreover, our clients have yet to seriously consider the advice of expressing breastmilk and leaving it at home. (PRC1, Health Center)*

Further, there was limited understanding of how to extract breast milk, which was perceived as unnatural unless the mother was hospitalized. This uncommon practice in the community resulted in a constraint on exclusive breastfeeding. There were also apprehensions about expressing and storing milk, sanitation, and the notion that it conflicted with God's established values. The participant said:

*I have tried to express breast milk several times, but it has never been suitable for my child. It feels like he gets more abdominal discomfort when I give them to him (P21, Health Center)*

*Early motherhood-related non-compliance to the safe infant feeding directive.* It was reported that most adolescent-age mothers were reluctant to follow the advice provided during the clinics and visits from CHWs. It was termed as ignoring the child's health outcomes, still feeling irresponsible for their child's health, and further deciding to adhere to their parents' and community's inexpert opinions on alternatives to breastfeeding. Stressing on this matter, the participant said:

*After realizing that I had a pregnancy and was diagnosed with HIV at 19, the home was no longer safe. So, I moved to live with my best friend (co-worker), who has been by my side for better or worse. I trust and share everything with her, even her opinion on feeding my child since she has two children and is more experienced (P22, RRH).*

The insight from a CHW revealed that knowledge uptake on infant and young child feeding was generally high among parents aged 24 and older. However, the CHW identified a challenge in getting younger parents to translate knowledge into practice, particularly in the 16–24 age group. While they may have verbally expressed willingness to follow health teachings, their child's growth did not consistently reflect this commitment. Instead, these younger parents often relied on peer opinions for guidance. The participant said:

*Knowledge uptake, in general, is high, but only above 24 years of age, mainly under that age, it has been challenging to uptake knowledge that we give to them. They will lie to follow our teachings, but it does not reflect on the child's progress, mostly being careless of their child's*

*growth; instead, they rely on peer opinions. The leading age group we struggle with is 16–24 years (PRC12, Health Center).*

This observation underscores the need for tailored approaches and interventions to effectively engage and educate younger parents on optimal infant and young child feeding practices.

*Postpartum depression.* Postpartum stress/depression fueled by the partner and mother's family emerged among the barriers that hinder mothers from practicing EBF, mainly reported an unhealthy relationship with their partners. Further, there is a lack of support for the different phases of motherhood from the partner aggravated with newly diagnosed HIV during the ante-natal clinic. As a result, giving up and neglecting self-health measures, including food intake, leads to a massive reduction in milk production and disinterest in breastfeeding the child. Describing this situation, a participant said:

*I would be lying to say I hate this child, but I would instead focus on my small business since no one has my back to provide for my other older child and me. I have been suffering a lot from pregnancy with no support, and now I must raise this child alone. I know the effect of alternative feeds, but I don't believe it is my fault alone. If I cannot produce enough milk, there is no way I will leave him hungry with no food (P16, Health Center).*

*Breast sores and abscess.* The health care provider reported sporadic cases, and for a few days (Five), as notified by one of the mothers interviewed, presented with pustules during the early days of breastfeeding, mostly treated and recovered within a short time with no recurrences. The participant said:

*In the second month of breastfeeding, my left breast developed something like an abscess. I immediately went to the clinic and was treated with medication and told to breastfeed on one side only and rest the affected breast while expressing the breast milk from it and dumping them. I recovered shortly without complications (P17, Health Center).*

**Theme 2: Household-level barriers to exclusive breastfeeding.** Participants also reported household-level barriers faced during exclusive breastfeeding practice. In this theme, three sub-themes, including food insecurity and inaccessibility to key resources, male partners and family members' influence on decision-making, and HIV status non-disclosure barriers to EBF support, are presented below. Verbatim quotations support the presentation.

*Food insecurity and inaccessibility to key resources.* Participants in this study revealed that their limited access to food was a primary reason for their inability to breastfeed their children exclusively. Due to insufficient meals (typically one to two per day), mothers produced inadequate breastmilk, which failed to satisfy their infants, and sometimes, they had no breastmilk at all. This food insecurity led them to seek help from relatives when possible, but maintaining consistent access to food for themselves and their infants remained challenging. Occasionally, this was reported, leading to considerations of alternative feeding practices.

*The main reason I failed to EBF my child was food, with one to a maximum of two meals per day leading me to produce very light breastmilk, which does not satisfy my child, and sometimes no breastmilk at all. So, I sometimes must go to my relatives to ask for food for its sake. However, I am not consistently successful; when I have no alternative for feeding myself, I consider something else for my child (P31, Health Center).*

Further, participants shared that the demanding nature of breastfeeding, coupled with the need to eat frequently to sustain the energy necessary for both breastfeeding and adhering to ART, constantly left them feeling weak. Despite their consistent efforts, the situation did not improve. Consequently, the participant made the difficult decision to cease breastfeeding altogether and transition to providing complete replacement feeds such as lactogen to their child.

*Breastfeeding was hard for me, and I was always weak. This is because I had to eat many times daily to maintain the energy required to breastfeed and ART intake, but the situation remains the same. Hence, I decided to stop for good and provide her with complete replacement feeds through lactogen, which significantly relieved my health and hers (P14, Health Center).*

These accounts highlight the intricate balance required for mothers living with HIV who are striving to breastfeed while managing their health through ART and the difficult decisions they often face.

A participant from a healthcare facility reported that a common concern among clients is poor nutrition, which affects a substantial portion of them. This nutritional deficiency often results in insufficient breast milk production, compelling many of these mothers to seek more affordable alternatives, with soft porridge being a common choice.

*Most of our clients face poor nutrition; I would say 6 out of 10 clients. That leads to insufficient breastmilk production and the choice of soft porridge as their most affordable alternative (PR2, RRH).*

*Male partners and family members have bold authority in decision-making*. In this sub-theme, most of the participants in the study reported taking a leading role in feeding their infants. While only a few said, their partners fulfilled that rule. Our participants presented the dominance of family members in making decisions on infants while ignoring or not seeking advice from the experts in health facilities. One of the participants said:

*I was not advised to feed my child from the clinic. I went there during pregnancy and after delivery; hence, I relied on my husband's advice. We fed him cow milk with soft cassava porridge when he was three months old. My child cannot feed on only breastmilk. Even those doctors' infants cannot sustain breastmilk alone, even without water! No way (P32, Health Center)*

Couples who were not willing to undergo HIV testing or visit the clinic showed a decreased tendency to value women's choices of exclusively breastfeeding their infants. In addition, most male partners seemed to lack vital information on their infants' proper feeding, hence improper influence on their wives.

*HIV-associated stigma, disclosure about their HIV status*. The findings revealed a high level of undisclosed HIV status among the co-parents and other close persons. A ratio of 4:10 of the participants had not disclosed their status to their partner. This negatively affects the practice of EBF despite the knowledge disseminated by the health care providers. HIV-positive mothers reported positive support from their spouses in ensuring food security for sufficient milk production and replacement food for those who opted to use complete replacement feeding (CRF) when becoming aware of their HIV status.

*My husband has been enormous support from the day I was diagnosed with HIV, during the clinic of this child. I lost hope and even wanted to abort. However, despite being tested*

*negative, he did increase his love and affection for me, and to date, he has been of hope and colossal support that I need in every way, from my health, feeding pattern, and protection for the child's health. We attend all the counseling sessions together and support each other to protect our child's health (P19, Health Center)*

Participants reported supporting their family members and partners' mixed feeding opinions to continue concealing their HIV status despite the special consideration required by their exposed children.

A notable trend has emerged in our research, as reported by CHW. Male partners who are aware of their wives' HIV status have demonstrated a commendable level of involvement. This heightened male-partner engagement is positively reflected in the practice of exclusive breastfeeding (EBF) among these couples. Additionally, it has translated into active participation in community outreach programs, underlining the importance of involving male partners in promoting EBF and broader community health initiatives.

*We have had an outstanding response from male partners who know their wives' status. It has been reflected in the EBF practice among those couples and significant participation in community outreaches (PRC12, Health Center).*

**Theme 3: Community-level barrier to exclusive breastfeeding.** Participants reported community-level barriers faced during exclusive breastfeeding practice. In this theme, only one sub-theme emerged, which is the Low retentivity of HIV-positive women in PMTCT programs. This is substantiated below with the support of a verbatim quotation.

*Low retentivity of HIV-positive women in PMTCT programs.* As reported by participants, keeping women and infants in PMTCT programs after delivery has been challenging. In addition, there has been gradual progress in HIV-positive mothers' detachment from postnatal clinics and community outreaches targeting PMTCT services. With a range of benefits to women and infants, one of the key supports of PMTCT services is safe childbirth and appropriate infant feeding. One of the healthcare providers said:

*As soon as they finish the cluster of vaccines given during the first three months, the number of HIV-positive mothers attending PMTCT clinics and community outreaches significantly reduces. You can imagine what comes next: reducing infant care and improper feeding. Unfortunately, most dropouts are those coming from the neighborhood, and they do so to conceal their status (PR1, Health Center)*

As a result, difficulty in reducing, detecting, and managing new infections among HIV-positive mothers throughout breastfeeding predisposes them to drop EBF practice. This observation underscores the need for tailored interventions and support to retain these mothers in PMTCT services and ensure the well-being of both mothers and infants.

## Discussion

This study explored the challenges HIV-positive mothers face in practicing exclusive breastfeeding (EBF) at different levels. It revealed individual, household, and community barriers that hinder the practice of exclusive breastfeeding (EBF) among HIV-positive mothers. Individual factors, such as occupation tie, early motherhood, postpartum depression, and breast conditions like sores and abscesses, were reported. Household challenges encompassed issues such as food insecurity, the impact of spouses and family authority on decision-making, and a

lack of substantial support from significant others due to HIV-related stigma. Notably, disclosure hesitance and HIV-related stigma emerged as intertwined factors in this complex scenario. Additionally, key informants highlighted a community-level barrier: the low retention of HIV-positive mothers in PMTCT programs.

The voice of our participants in quotes from the results section sheds light on the critical issue of occupation as a significant barrier to exclusive breastfeeding (EBF) among HIV-positive mothers. HIV-positive mothers are often required to balance their occupational responsibilities with childcare, thus encountering substantial hurdles in ensuring EBF. The demands of the workplace can frequently disrupt breastfeeding schedules, leading to suboptimal feeding practices. This observation aligns with the broader literature on working mothers' challenges sustaining EBF. Employment can positively impact breastfeeding as it provides financial stability and a sense of purpose, allowing mothers to invest in their health and the health of their infants [41].

On the contrary, employment can also negatively affect breastfeeding, as mothers reported being unable to breastfeed exclusively due to work schedules, lack of appropriate facilities, or stigma surrounding HIV and breastfeeding, similar to other contexts [42]. Employed women are more likely to practice EBF than unemployed women [43]. Some of the reasons, such as employment-related stress and lack of flexible working hours, are reported to be associated with lower rates of EBF among HIV-positive mothers [44]. Providing workplace support, including designated spaces for breastfeeding, lactation breaks, and education on the benefits of exclusive breastfeeding, has the potential to increase exclusive breastfeeding rates among working mothers and can also positively impact the exclusive breastfeeding rates among HIV-positive mothers [45, 46].

The age of mothers plays a crucial role in their ability to adhere to exclusive breastfeeding (EBF) practices, particularly among HIV-positive mothers. Despite high knowledge uptake on EBF as reported by healthcare providers, a notable distinction arises when considering age groups. Mothers aged 24 and above tend to have a more significant understanding and commitment to EBF, while those under this age, especially teenagers, face challenges in adopting the knowledge provided. Adolescent-aged mothers were pessimistic towards EBF due to the peer influence from their age mates and parents, making it difficult for them to initiate and sustain exclusive breastfeeding, similar to other contexts [11]. Mothers often report being advised by their families to use mixed feeding for their infants, even if they disagree, primarily if they rely on their families for support [47]. Adolescent mothers who are HIV positive might lack experience and confidence in their beliefs, which could lead them to seek assistance with parenting from their family members, especially their mothers and grandmothers [42]. This is particularly true for adolescent mothers who are HIV-positive and may feel unsure about their parenting abilities [42]. They may turn to their mothers and grandmothers for guidance, and accommodating their families' wishes may help them cope with the demands of parenting while still developing themselves [47]. However, there is inconsistency in the findings of different studies. For example, younger mothers had higher exclusive breastfeeding rates than older mothers, as reported elsewhere, and HIV-positive mothers under 25 had higher exclusive breastfeeding rates than those over 25 [43, 48]. It is, therefore, essential to note that a mother's age is not the sole determinant of exclusive breastfeeding practices [49]. Other factors, including access to healthcare services, education, and social support, can also influence exclusive breastfeeding rates [6, 49].

This study demonstrates that mothers who experience emotional breakdowns and stress are at a higher risk for poor self-efficacy and may neglect their infants' care. Unhealthy family relationships, loneliness, and feelings of rejection can lead to severe stress and depression. Thus, mothers who lack psychosocial support may struggle with the challenges associated with

exclusive breastfeeding (EBF) and experience guilt and feelings of inadequacy. Numerous studies have indicated that social and family support quality can positively impact neuroendocrine functioning and mood [50–54]. These emotions are often linked to early discontinuation of breastfeeding and exacerbate the connection between maternal stress and psychological distress [55, 56]. Prenatal depression also significantly affects EBF rates, with studies reporting poor rates in South Africa and Australia [57]. Therefore, the presence of depression during pregnancy in women with HIV poses a risk to their capacity to effectively follow the World Health Organization's guidelines, which include exclusive breastfeeding for up to 6 months and adherence to antiretroviral therapy. A significant number of self-reported depression within six months post-delivery reported in Australia pose more threat to EBF discontinuation and hence require more attention [49]. Women with more depressive symptoms were less likely to initiate breastfeeding exclusively, as noted in previous studies of depressive symptoms and poor infant nutrition [58]. Further, women with depressive symptoms are less likely to start and maintain exclusive breastfeeding, as seen in studies from Kenya and other countries [57, 59].

Breast conditions, such as abscesses, mastitis, or other infections, pose a considerable threat. In response, healthcare providers often recommend limiting breastfeeding from the affected breast while expressing and discarding the milk to prevent potential transmission of the virus. While such interventions can help mitigate risks, they introduce additional burdens and complexities for mothers. HIV-positive mothers are advised to stop breastfeeding and seek immediate treatment while using alternative feeds that are safe for their infants' such as safe formula feed [60, 61]. However, the cost and the competitive price make it difficult for them to sustain for a slightly extended period, hence including cow milk among the options [62]. HIV infections create a significant risk of developing breast infection [63], and the risk is reduced with the effective use and adherence to potent ART. The same findings were reported in Ghana, where some participants identified improper positioning of the baby as the cause of sore nipples [38]. In addition, incorrect techniques, less frequent breastfeeding, scheduled times, and pacifiers were reported to predispose to lactation problems [64]. Breast-related conditions are preventable if the mother empties her breasts and effectively starts post-delivery immediately and beyond. Properly managing these conditions is crucial; if not treated, it might lead to early weaning.

The study findings illustrate that mothers facing food insecurity often struggle to provide sufficient and nutritious breastmilk to satisfy their infants. The struggle to nourish their infants while grappling with their hunger adds a layer of stress to their lives. This emotional burden can negatively impact the mother's well-being, potentially affecting her capacity to engage in EBF. Inadequate nutrition can have a notable impact on the physiology of the breast and the production, secretion, and composition of milk [65]. In addition, the infant's fussiness and poor weight gain force them to supplement their breast milk with other food sources and, as a result, reduce the benefits of exclusive breastfeeding for both the mother and the baby. Similar associations between food insecurity and negative attitudes toward EBF have been found in other contexts [44, 45]. In Malawi, food insecurity was a significant factor in HIV-affected women's decisions to continue breastfeeding [10]. Women often believe inadequate diets lead to insufficient milk production, increasing the likelihood of supplementing their infants' [13, 42, 66]. Their hunger drives the perception of their body's inability to produce sufficient milk, reinforcing mixed feeding to meet their perceived needs for their children [67].

The social dynamics surrounding infant feeding have a considerable impact on a woman's ability to maintain optimal breastfeeding practices, regardless of her HIV status. While healthcare providers offer guidance to address household obstacles, such as pre-lacteal feeds or herbal remedies and water, mothers-in-law, and husbands, who play an essential role in supporting mothers in the domestic sphere, often provide erroneous advice regarding infant feeding. Consequently, their understanding of infant feeding becomes the primary source of

information for women in this community. This study is in line with the systematic review of the views, attitudes, and behaviors of both individuals and societies concerning mother-to-child transmission and infant feeding among HIV-positive mothers in SSA [68], demonstrating that household culture and social norms influence a mother's decision to EBF [68]. In addition, interventions encouraging male partner involvement, such as enhanced psychosocial interventions, verbal encouragement, and complex community interventions, increase safe infant feeding practices.

The act of revealing one's HIV status is a complicated matter since it carries crucial psychological, social, cultural, and financial consequences. In this study, participants who reported not disclosing their HIV status to anyone, including their partners, were most likely to favor mixed feeding practices. In contrast, women who declared their status reported massive support from their close people, which influences EBF practice and collective support for infant feeding. Healthcare providers also commented on the positive impact of the disclosure among the couple that disclosed to each other on adherence to the infant feeding recommendations and enabling her to take medication without fear. This finding agreed with the study conducted in Nigeria [69]. Further, EBF practices among HIV-positive mothers who disclosed their status to their spouses were nearly six times more likely than mothers who did not disclose their status in Ethiopia [70]. The possible reason would be that if mothers declare their HIV status to their spouses, they might get good family care. For instance, some reasons for disclosure were decreased workload, nutritional reinforcement, compliance with ART treatments, and exclusive breastfeeding motivation [13, 71]. Moreover, disclosing HIV status to the spouse prevents mixed feeding [70]. This is because HIV-positive mothers who tell their status to their partner will get adequate care, support, and time for breastfeeding.

Poor retention of HIV-positive mothers to PMTCT services, which starts from the Antenatal clinic (ANC), was also reported to be associated with reduced EBF practice in our study despite the local availability of health facilities. The main reason for poor clinic attendance was women's desperation to conceal their HIV status from close family members. This compels them to stay away from the facilities where HIV status was diagnosed, where they can receive PMTCT services, and attend perinatal clinics where status is unknown. Several studies have reported the positive contribution of PMTCT services to EBF practice among HIV-positive mothers [72–74]. As proof, PMTCT services improved knowledge, attitudes, and exclusive breastfeeding practices among HIV-positive mothers [72–74]. Further, PMTCT services help educate and support mothers to practice exclusive breastfeeding which is crucial in reducing the risk of mother-to-child virus transmission and providing essential nutrients to the infant [73]. The challenge for women to disclose their serostatus and the stigma associated with it creates obstacles to their engagement in PMTCT programs and starting or continuing ARV therapy for their well-being and that of their children [75, 76]. Consequently, they are less inclined to adopt exclusive breastfeeding practices [75, 76]. However, women are concerned about the implication of close relatives becoming aware of their positive status. Augmenting essential PMTCT services with mentor mothers and a culturally adapted cognitive behavioral intervention helped give information and improve the emotional perspective of HIV-positive pregnant women [77]. Hence, healthcare providers should mount efforts to enhance the effectiveness of PMTCT programs and identify and involve these significant others to help disperse the stigmatization of HIV-positive women.

## Study limitations and strengths

The study acknowledged social desirability bias in discussing infant feeding practices in healthcare settings. To mitigate this, the study maintained transparency by clearly stating its

objectives before interviews. Participants were assured that there would be no adverse consequences related to their chosen feeding methods, promoting an open environment during the research. Furthermore, using Kiswahili, the national language, during data collection facilitated open and candid communication, allowing participants to express themselves comfortably in their native language, which aided in code identification. Additionally, to enhance the trustworthiness of our qualitative research, we employed data triangulation involving three distinct participant groups and utilized two researchers for transcript coding. Including direct quotes from participants describing their experiences further enhances the reliability of the study findings, providing readers with concrete evidence of the study's dependability.

## Conclusions and recommendations

This study sheds light on the intricacies of optimal breastfeeding, specifically exclusive breastfeeding as a method of infant feeding for HIV-positive mothers. The results indicated several barriers to exclusive breastfeeding (EBF) among these mothers. Individual factors, such as a busy work schedule, early motherhood, postpartum depression, and breast conditions such as sores and abscesses, were significant challenges to EBF. Household challenges, including food insecurity, the influence of spouses and family authority on decision-making, and lack of substantial other support due to HIV-related stigma and disclosure hesitance, hindered EBF practice. HIV-related stigma and disclosure hesitance were also barriers to EBF. Furthermore, low retention of HIV-positive mothers in PMTCT programs emerged as a significant obstacle to EBF practice in the community. These findings suggest a high need for a more individualized, household, and communalized approach to sustaining HIV-positive mothers to adhere to exclusive breastfeeding.

## Supporting information

**S1 File. Interview guide.**
(PDF)

## Acknowledgments

We extend our sincere gratitude to all participants whose consent to participate was integral to the success of this study. Special appreciation is extended to Harvard T. H Chan and Muhimbili University of Health and Allied Science for sponsoring the data collection part of this study through their collaborative project, the HIV Implementation Science Grant and NIH D43 grant (1D43TW009775-01A1). Additionally, we express profound thanks to Amana Regional Referral Hospital, Mnazi Mmoja, and Buguruni Health Centres for accepting our study to be conducted at the Reproductive and Child Health (RCH) clinic and for providing a supportive environment throughout the data collection process. Lastly, we acknowledge with gratitude all individuals who contributed in various capacities to the success of this study.

## Author Contributions

**Conceptualization:** Goodluck Augustino, Amani Anaeli, Bruno F. Sunguya.

**Data curation:** Amani Anaeli, Bruno F. Sunguya.

**Formal analysis:** Goodluck Augustino, Amani Anaeli, Bruno F. Sunguya.

**Funding acquisition:** Bruno F. Sunguya.

**Investigation:** Goodluck Augustino.

**Methodology:** Goodluck Augustino, Amani Anaeli.

**Supervision:** Bruno F. Sunguya.

**Writing – original draft:** Goodluck Augustino.

**Writing – review & editing:** Amani Anaeli, Bruno F. Sunguya.

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
