## [Decision Letter · Decision Letter 0]

8 Jan 2024

PONE-D-23-42097Infant and Young Child Feeding in the Context of HIV: An Exploration of Barriers in Exclusive Breastfeeding Practice in Dar es Salaam, Tanzania.PLOS ONE

Dear Dr. Augustino,

Thank you for submitting your manuscript to PLOS ONE. After careful consideration, we feel that it has merit but does not fully meet PLOS ONE’s publication criteria as it currently stands. Therefore, we invite you to submit a revised version of the manuscript that addresses the points raised during the review process.

We look forward to receiving your revised manuscript.

Kind regards,

Kahsu Gebrekidan

Academic Editor

PLOS ONE

Journal Requirements:

3. Thank you for stating the following financial disclosure: "Data collection was supported by funding from the HIV Implementation Science Grant, a collaborative NIH D43 grant (1D43TW009775-01A1) by Harvard T.H. Chan School of Public Health and MUHAS"

Reviewers' comments:

Reviewer's Responses to Questions

**Comments to the Author**

1. Is the manuscript technically sound, and do the data support the conclusions?

Reviewer #1: Yes

Reviewer #2: Yes

2. Has the statistical analysis been performed appropriately and rigorously? 

Reviewer #1: N/A

Reviewer #2: N/A

3. Have the authors made all data underlying the findings in their manuscript fully available?

Reviewer #1: Yes

Reviewer #2: Yes

4. Is the manuscript presented in an intelligible fashion and written in standard English?

Reviewer #1: Yes

Reviewer #2: Yes

5. Review Comments to the Author

Reviewer #1: Thank you for generating relevant evidence on exclusive breast feeding in the context of HIV. Generally, it is well written. However, to improve the readability of the article, consider the comments described below.

Title

The entire manuscript discusses about exclusive breast feeding which is one fore of IYCF. So, the current title is too broad. Limit it to exclusive breast-feeding practices in HIV positive women. And a clear title should include the problem, the subjects/study participants and the place. Consider this in your title too.

Methods

1. Data collection

Description of who conducted the interview with the different participant groups and the interviews skill of collecting interview data (qualitative research) and what tools (audio recorder) were used to capture the interview with participants will be important to get clarity about your data collection. Include this information. How did you do verbatim transaction of each interview from field notes?

2. You have collected socio-demographic data to ensure participants from different socio demographic groups are represented in the interviews, which is good. If so, present the summary of participants characteristics in table.

Result

1. Treatment of different views

Do all participant groups have similar view in each theme and sub theme described? If there are deviants, their view should be reflected.

2. Triangulation

How is the data from health professionals triangulated with finding from HIV positive women? Can you explain a bit further?

Good luck with your revision!

Reviewer #2: Title: Infant and Young Child Feeding in the Context of HIV: An Exploration of Barriers in Exclusive Breastfeeding Practice in Dar es Salaam, Tanzania

General comments

• The title is fantastic and has brought some important findings. However, the study populations should be mentioned in the title.

• Please use the guidelines of the consolidated checklist for reporting in-depth interviews and focus group discussions in qualitative research. In my opinion, the current document lacks flow, especially in following the standard of qualitative research. The methodology needs a detailed description. Otherwise, well done!

• In abstract part, the methods part should contain study design, population, sample, sampling method, data collection, and type of data, tool, data analysis, and the way you interpreted. Please avoid traditional narration since qualitative research is “live” research. In my opinion, it is better to show the clear direction of exploring data from each source chronologically so that readers also understand where the insight information is documented and the method of exploration.

• In abstract part: findings, you wrote, “The study identified various barriers to exclusive breastfeeding, encompassing individual factors." Please avoid introducing results to the word economy abstract; rather, just write the main findings and their link with study populations. Make the reader feel motivated to say the findings are live.

• In abstract part, Conclusion and Recommendations, I haven’t seen what you wrote here, in the findings. Theoretically, recommendations should emanate from results.

• Introduction: Well written. However, since this is a qualitative study, you should analyze the individual, community, and health care-related factors beyond the numbers. Otherwise, it is fantastic.

• In Methods: As you can see from your method, there is no specific design explicitly written here. As to me, there are a variety of designs to be used in a qualitative study; please indicate the appropriate ones.

• In Methods: Data Collection and Tools, in-depth interviews and key informant interviews should be merged under the heading “Data collection methods and tools” and should be procedurally described.

• In Methods: Data Management and Analysis Strategies is very well described. However, this part should need a detailed description of the “trustworthiness of qualitative research." Which is missed!

• In result part, I found it very nice. However, conceptualize the whole concept of the emerging themes and anomalies in the beginning so that readers can be interested.

• Table 1 should be formatted and auto-fit to the window. Otherwise, it is fantastic.

• This is a new view of the result or study; perhaps I am expecting from you to include the relationships between the variables obtained from your result in network or diagram form in order to build the idea. Since in qualitative research, all is data.

• In discussion part, it was well narrated and discussed.

• Putting honest limitations first is best.

• References: Please make sure that all the references are used in the document and updated.

I am happy to review this fantastic and well-written paper.

6. PLOS authors have the option to publish the peer review history of their article (what does this mean?). If published, this will include your full peer review and any attached files.

Reviewer #1: No

Reviewer #2: No

---

## [Author Response · Author response to Decision Letter 0]

23 Feb 2024

We also appreciate the time and effort you and each reviewer have dedicated to providing insightful feedback on ways to strengthen our paper. Thus, it is with great pleasure that we resubmit our article for further consideration. We have incorporated changes that reflect the detailed suggestions you have graciously provided. We hope our edits and responses below satisfactorily address all the issues and concerns you and the reviewers have noted.

To facilitate your review of our revisions, we have attached rebuttal letter with a point-by-point response to the questions and comments in your letter dated Jan 8, 2024

---

## [Decision Letter · Decision Letter 1]

13 May 2024

Barriers to exclusive breastfeeding practice among HIV-positive mothers in Tanzania. An exploratory qualitative study.

PONE-D-23-42097R1

Dear Mr Goodluck,

We’re pleased to inform you that your manuscript has been judged scientifically suitable for publication and will be formally accepted for publication once it meets all outstanding technical requirements.

Kind regards,

Kahsu Gebrekidan

Academic Editor

PLOS ONE

Additional Editor Comments (optional):

Reviewers' comments:

Reviewer's Responses to Questions

**Comments to the Author**

1. If the authors have adequately addressed your comments raised in a previous round of review and you feel that this manuscript is now acceptable for publication, you may indicate that here to bypass the “Comments to the Author” section, enter your conflict of interest statement in the “Confidential to Editor” section, and submit your "Accept" recommendation.

Reviewer #1: All comments have been addressed

Reviewer #2: All comments have been addressed

2. Is the manuscript technically sound, and do the data support the conclusions?

Reviewer #1: Yes

Reviewer #2: Yes

3. Has the statistical analysis been performed appropriately and rigorously? 

Reviewer #1: N/A

Reviewer #2: Yes

4. Have the authors made all data underlying the findings in their manuscript fully available?

Reviewer #1: Yes

Reviewer #2: Yes

5. Is the manuscript presented in an intelligible fashion and written in standard English?

Reviewer #1: Yes

Reviewer #2: Yes

6. Review Comments to the Author

Reviewer #1: Thanks dear authors,

The article is well written and the findings are useful for scientific and practice community.

At this point, I have 4 comments to be addressed.

1. Adequate explanation about the study design is required. I.e. which aspect/characteristics makes the study design a "case study". It could be just exploratory qualitative study or provide further explanations that it is a case study.

2. How did you ensure the diversity of sub-categories in the data. Why is the term adolescents mentioned on page 4?

3. Labeling the title for tables needs improvement (what, where, and when).

4. Where is the application of integrated model of behavioral prediction in this article.

Reviewer #2: All the comments have been addressed. Thank you for your response dear authors. Well responded and addressed.

7. PLOS authors have the option to publish the peer review history of their article (what does this mean?). If published, this will include your full peer review and any attached files.

Reviewer #1: No

Reviewer #2: No

---

## [Editor Report · Acceptance letter]

17 May 2024

PONE-D-23-42097R1 

PLOS ONE

Dear Dr. Augustino, 

I'm pleased to inform you that your manuscript has been deemed suitable for publication in PLOS ONE. Congratulations! Your manuscript is now being handed over to our production team.

Kind regards, 

on behalf of

Dr. Kahsu Gebrekidan 

Academic Editor

PLOS ONE